

# The impact of ancestral heath management on soils and landscapes: a reconstruction based on paleoecological analyses of soil records in the middle and southeast Netherlands

**M. Doorenbosch**[1] **and J. M. van Mourik**[2]

[1]Faculty of Archaeology, University of Leiden, Einsteinweg 2, 2333CC Leiden, the Netherlands
[2]Institute for Biodiversity and Ecosystem Dynamics, University of Amsterdam, Science Park 904, 1098 XH Amsterdam, the Netherlands

Received: 14 November 2015 – Accepted: 4 December 2015 – Published: 18 January 2016

Correspondence to: M. Doorenbosch (m.doorenbosch@arch.leidenuniv.nl)

Published by Copernicus Publications on behalf of the European Geosciences Union.

SOILD

doi:10.5194/soil-2015-83

The impact of ancestral heath management on soils and landscapes

M. Doorenbosch and J. M. van Mourik

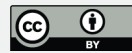

**SOILD**

doi:10.5194/soil-2015-83

**The impact of ancestral heath management on soils and landscapes**

M. Doorenbosch and J. M. van Mourik

## Abstract

The evolution of heath lands during the Holocene has been registered in various soil records. Paleoecological analyses of these records enable to reconstruct the changing economic and cultural management of heaths and the consequences for landscape
and soils.

Heaths are characteristic components of cultural landscape mosaics on sandy soils in the Netherlands. The natural habitat of heather species was moorland. At first, natural events like forest fires and storms caused small-scale forest degradation, in addition on the forest degradation accelerated due to cultural activities like forest
grazing, wood cutting and shifting cultivation. Heather plants invaded on degraded forest soils and heaths developed. People learned to use the heaths for economic and cultural purposes. The impact of the heath management on landscape and soils was registered in soil records of barrows, drift sand sequences and plaggic Anthrosols. Based on pollen diagrams of such records we could reconstruct that heaths were
developed and used for cattle grazing before the Bronze Age. During the Late Neolithic, the Bronze Age and Iron Age, people created the barrow landscape on the ancestral heaths. After the Iron Age people probably continued with cattle grazing on the heaths and plaggic agriculture until the Early Middle Ages. After AD 1000 two events affected the heaths. At first deforestation for the sale of wood resulted in the first regional
extension of sand drifting and heath degradation. After that the introduction of the deep stable economy and heath sods digging resulted in acceleration of the rise of plaggic horizons, severe heath degradation and the second extension of sand drifting. At the end of the 19th century the heath lost its economic value due to the introduction of chemical fertilizers. The heaths were transformed into "new" arable fields and forests
and due to deep ploughing most soil archives were destroyed. Since AD 1980, the remaining relics of the ancestral heaths are preserved and restored in the frame of the programs to improve the regional and national geo-biodiversity.

Title Page

Abstract | Introduction

Conclusions | References

Tables | Figures

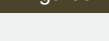

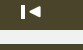 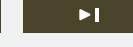

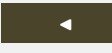 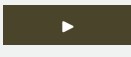

Back 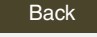 | Close 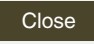

# 1 Introduction

Heaths are characteristic biotopes in Northwestern Europe. Most of the heaths in the study area occur on coversand, deposited during the Late Glacial and Preboreal. The current biodiversity management regards dry and moist heaths as region specific ecotopes that need to be preserved and protected (LNV, 2001, 2006). *Erica tetralix* dominated the moist, *Calluna vulgaris* the dry heaths; both species grow on acid, humic soils, poor in Nitrogen, phosphate and carbonate (Weeda et al., 1988; IVN, 2001). Originally, the expression heath was not connected to a vegetation type, but to common fields for cattle grazing on chemically poor sandy soils. The family of Ericaceae counts ≈ 3000 cosmopolitan species ligneous plants of which seven occur in the Netherlands. Two factors are crucial for the survival of heaths on chemical poor sandy substrates (Weeda et al., 1988).

1. The heather plants live in symbiosis with *ericoide mycorriza*. These organisms exclusively provide the heath plants with nitrogen that is not available for other plants, living without this form of symbiosis.

2. Heath plants have specific adaptations to drought stress. Leaves will stop evaporation of water under dry or warm air conditions, but can evaporate big amounts of water under humid air conditions. This is necessary to concentrate enough nutrients from soil water with very low nutrient concentrations for the plant growth.

The natural habitats for heaths were the moors on poorly drained sandy soils. *Erica tetralix* dominated in the lowness of the moors, *Calluna vulgaris* on the dryer rises of *Sphagnum* peat (Weeda et al., 1988).

The oldest occurrence of Ericaceae in the coversand landscape was reported in pollen diagrams of initial Histosols, developed during the Bølling and Allerød interstadials (van Geel et al., 1989; van Mourik and Slotboom, 1995). The Holocene migration of heather species from the moors to the coversand landscapes was initially

Discussion Paper | Discussion Paper | Discussion Paper | Discussion Paper |

**SOILD**

doi:10.5194/soil-2015-83

**The impact of ancestral heath management on soils and landscapes**

M. Doorenbosch and J. M. van Mourik

Title Page

Abstract | Introduction

Conclusions | References

Tables | Figures

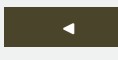 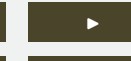

triggered by environmental events as storm and forest fires, but accelerated by cultural factors as deforestation and agriculture.

The Holocene vegetation development and soil formation on coversands started without any human interference and continued until $\approx 3000$ BC. At that time, the soilscape was in accordance with the geomorphological structure, xeromorphic Podzols on coversand ridges, gleyic Podzols on coversand planes, histic Podzols in brook valleys. From the Late Neolithic on, the effect of human land use on soil and landscape increased. The deciduous forest degraded by woodcutting, forest grazing and shifting cultivation (van Mourik et al., 2012, 2013). *Calluna* entered the area and the heaths expanded.

Nowadays, moist and dry heaths are valuable naturel target types in the Netherlands, as confirmed in reports about the future designs of the national and European ecological master structure (LNV, 2001, 2006). To preserve the heaths, future sustainable heath management must be based on knowledge of the origin of heath biotopes and the role of heaths in historical land use systems (Smits and Noordijk, 2013).

Information about historical land management on coversands can be unlocked from soil archives, found in barrow and driftsand landscapes and plaggic Anthrosols. Unfortunately, in the course of the 20th century, the majority of the heaths were transformed into arable land and forest plantations, soils were deeply ploughed and soils archives were severally damaged or even destroyed (Fig. 1) and the total heath surface decreased from 600 000 to 30 000 ha (Fig. 2).

Remaining parts of eligible soil archives were palaeosols in barrow landscapes (Doorenbosch, 2013), polycyclic sequences in drift sand landscapes (van Mourik et al., 2012, 2013) and plaggic Anthrosols (Spek, 2004; van Mourik et al., 2011). A part of these archives is now included in managed nature reserves and sustainable protected as part of the cultural heritage.

The aim of this paper is to reconstruct the impact of ancestral heaths management on soil and landscape.

**SOILD**

doi:10.5194/soil-2015-83

**The impact of ancestral heath management on soils and landscapes**

M. Doorenbosch and J. M. van Mourik

The soil records, found in the barrow landscapes (Doorenbosch, 2013) cover the period between ≈ 3000 BC and the Roman Time, records found in driftsand landscapes and on plaggic Anthrosols cover the period between AD 500 and 1900. Soil records focusing on heath management from the beginning of the Roman Time until AD 500

were not found, but the continuity of the Ericaceae in pollen diagrams of moors and palaeosols, suggest the continuity of heaths during these period, probably as extensive grazing areas, without clear registration in the soil archives.

## 2 Materials and methods

### 2.1 Profile selection

Barrows have been built from ≈ 3000 BC (the late Neolithic period) until ≈ 100 BC (the Late Iron Age). Around 4000 barrows are known to be still present in the Netherlands, but considering the large amount of barrows that has disappeared over time there must have been thousands of burial mounds in the Netherlands, dominating the morphology of the landscape.

Many of these burial mounds have been excavated and sampled for pollen analysis to reconstruct the barrow environment (Fig. 3). To reconstruct environmental development, pollen spectra of samples of the mounds are inferior, but spectra of samples of the buried Podzolsl underneath a barrow, with on top the pre barrow land surface are opportune with the restriction of the regular complications of soil pollen

spectra (van Mourik, 2001). Pollen grains precipitate onto the land surface and infiltrate by bioturbation into the soil profile and reach the A, E, B and even the C horizons. Soil acidification, a regular development during the Holocene in sandy substrates, caused retrogressive activity during acid soil formation. Hence, older pollen assemblages will be preserved in the lower parts of the soil (van Mourik, 1999, 2001). When the soil

was buried by the construction of the barrow, the active soil processes stopped and the

**SOILD**

doi:10.5194/soil-2015-83

**The impact of ancestral heath management on soils and landscapes**

M. Doorenbosch and J. M. van Mourik

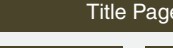

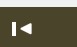 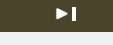 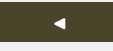 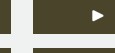

soil record was conserved. The pollen grains, incorporated in the palaeosol before and during the time that barrow was built are nowadays in many cases still present.

A barrow is usually constructed of sods. These sods were taken from the upper part of the soil in the surroundings and placed upside down when building the barrow. Sods contain parts of the soil record from the place where they were taken, including pollen grains. Pollen spectra of the constructing materials and the palaeosol have been used to reconstruct the morphology of the barrow landscape (Waterbolk, 1954; Groenman-van Waateringe, 1988; Bloemers, 1988; Doorenbosch, 2013). These investigations have revealed that all studied barrows were built in heath vegetation that has been managed for several millennia.

For this paper we used the cases Renkum stream valley, Vaassen-Niersen and Echoput (Doorenbosch, 2013) to demonstrate the heath management of the barrow landscape between ≈ 3000 and ≈ 200 BC. Heath management after AD 500 caused two changes in the geomorphology of the landscape. Firstly the plaggic horizons developed and the surface of the arable fields raised, secondly severe heath degradation resulted in sand erosion and redisposition. We used the case study Bedafse Bergen (van Mourik, 2013) to demonstrate the impact of heath management on soils and landforms between AD 500 and 1900.

## 2.2 Pollen analysis

Pollen records of palaeosols in barrow landscapes, buried podzols and Anthrosols provide paleoecological information about plant species, present on site and in the region during the formation of the barrows, driftsand deposits and plaggic horizon. Previous research showed that pollen grains, infiltrated in soils and incorporated in plaggic deposits, are well preserved in the anaerobic and acid microenvironment of excremental aggregates (van Mourik, 1999, 2001).

Samples for pollen extraction were collected in 10 mL tubes in profile pits. For a correct matching of pollen and biomarker spectra of the plaggic deposits, the same samples were treated for both pollen and biomarker extraction and analysis. The pollen

**SOILD**

doi:10.5194/soil-2015-83

**The impact of ancestral heath management on soils and landscapes**

M. Doorenbosch and J. M. van Mourik

Discussion Paper | Discussion Paper | Discussion Paper | Discussion Paper

extractions were carried out using the tufa extraction method (Moore et al., 1991, p. 50). For the identification of pollen grains, the pollen key of Moore et al. (1991, p. 83–166) and Beug (2004) was applied. The pollen scores of the barrow records are based on a tree pollen sum minus *Betula* (in the curve of total AP [= arboreal pollen] *Betula* is included). Pollen scores of the archives of Bedafse Bergen were based on a total pollen sum of arboreal and non-arboreal plant species. For the estimation of the pollen concentrations of the various soil horizons of profile Rakt, the exotic marker grain method was applied (Moore et al., 1991, p. 53).

## 3  Results

### 3.1  The Renkum stream valley

In a stream valley near Renkum (for a location map see Fig. 2) an alignment of barrows is situated with a length of at least 4.5 km. Several barrows of this alignment have been excavated and sampled for pollen. Pollen samples were taken from the old surfaces underneath the barrows and/or the sods the barrows were built of. Palynological analyses were performed by Casparie and Groenman-van Waateringe (1980, p. 24–36), with the exception of Bennekom 1. Bennekom 1 was published by van Giffen (1954). Doorenbosch (2013) reinterpreted and published the data retrieved by the above-mentioned researchers in her PhD thesis. The barrows were dated to the late Neolithic A ($\approx$ 2900–2500 BC) and the late Neolithic B period ($\approx$ 2500–2000 BC).

Figure 4 shows the pollen spectra from the barrows of the alignment in the Renumber stream valley. In the Neolithic A period pollen from heath form a considerable part of the pollen spectra. Heath pollen tends to spread only within a few meters from the place the heath is growing and the pollen is produced. This implies that the considerable percentage of heath pollen indicates that all investigated barrows in the area were constructed in an open space with vegetation where heath was an important component. In addition to heath grasses also formed part of the vegetation in the

**SOILD**

doi:10.5194/soil-2015-83

**The impact of ancestral heath management on soils and landscapes**

M. Doorenbosch and J. M. van Mourik

open places. Arboreal pollen percentages fluctuate between barrows from around 45 to around 75 %, indicating varying sizes of the open spaces the barrows were situated in. Based on research that was performed in recent heath areas with varying distances to the forest, such arboreal pollen percentages indicate that the open spaces had an average distance to the forest from 30 until 250 m (Doorenbosch, 2013, chapter 7). The forest consisted mainly of oak (*Quercus*) and hazel (*Corylus*), while alder carr (*Alnus*) was present in the wetter parts of the landscape. The barrows that date to the late Neolithic B period show a similar vegetation composition. Apparently these barrows were also built in open spaces, where heath and grasses were the main components. The size of the open spaces seems to be smaller than during the Neolithic A period. Arboreal percentages are lower, indicating an ADF (average distance to the forest) of approximately 50 m.

Besides the barrows that have been palynogically investigated in this area, palynological data from many other barrows on the Pleistocene coversand areas in the Netherlands are known (see Sects. 3.2, 3.3; see also Doorenbosch, 2013). These data show that these barrows were also built in heaths. The amount of barrows in the investigated regions was much more numerous than the amount of palynologically analysed barrows. In addition, the original number of barrows was probably even higher, since only a fraction of barrows has been preserved over the last millennia (Bourgeois, 2013, p. 40). Considering that all investigated barrows were built in heath vegetation it is probable that the non-investigated barrows in these areas were also built in open spaces where heath vegetation was dominant. The barrows of Renkum were built in an alignment and the distance between the barrows is mostly within a few hundred meters. Since the *average* distance to the forest of the open spaces varies from 50 to 300 m, it is likely that the open spaces were connected to each other, forming relatively small but long-stretched heathland areas with of length that could add up to several kilometres (Fig. 5).

**SOILD**

doi:10.5194/soil-2015-83

**The impact of ancestral heath management on soils and landscapes**

M. Doorenbosch and J. M. van Mourik

## 3.2 Vaassen–Niersen

A second example is given in Fig. 7, showing the results of several barrows that are situated in the northeastern part of the Veluwe (for the location see Fig. 2). In this area several barrows alignments and solitary barrows are present. Palynological data from the old surfaces and sods from several barrow periods and from ditches associated with the barrows are available for five barrows in this area, dating from the late Neolithic A to the Middle Bronze Age period ($\approx$ 1800–1100 BC) (Casparie and Groenman-van Waateringe, 1980; Doorenbosch, 2013). Two of these barrows form part of a barrow alignment.

The pollen spectra show that the barrows in this area, as the barrows in the Renkum stream valley, were built in open places with heath vegetation, surrounded by oak forest and alder carr in the lower parts of the area. The open spaces were probably larger than in the Renkum stream valley, with an ADF of around 100 m for the barrows of Vaassen and an ADF of 100–200 m for the barrows of Niersen. The vegetation of the open space seems stable, since the barrow spectra from all represented periods show similar vegetation patterns: an open place with species-poor grassy heathland surrounded by oak forest with an alder carr nearby. Figure 8 shows the visual impact on the landscape in the area of Vaassen–Niersen, assuming all barrows were built in heath.

## 3.3 Echoput

A third example concerns the twin barrows of the Echoput (site indicated in Fig. 2), which date to the Middle or earlier Late Iron Age: mound 1: 331–203 cal BC terminus post quem, 376–171 cal BC terminus ante quem; mound 2: 326–204 cal BC terminus post quem (Bourgeois and Fontijn, 2011, p. 87; Doorenbosch, 2013; van der Linde and Fontijn, 2011, p. 62). These barrows were excavated and sampled for pollen analyses by the Archaeology department of the University of Leiden (Doorenbosch, 2011). The pollen analyses were performed by the first author of this article as part of her PhD research (Doorenbosch, 2013). From both burial mounds samples were taken from

Discussion Paper | Discussion Paper | Discussion Paper | Discussion Paper |

**SOILD**

doi:10.5194/soil-2015-83

**The impact of ancestral heath management on soils and landscapes**

M. Doorenbosch and J. M. van Mourik

Title Page

Abstract | Introduction

Conclusions | References

Tables | Figures

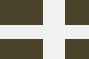 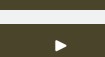

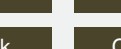

Back | Close

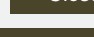

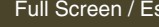

**SOILD**

doi:10.5194/soil-2015-83

**The impact of ancestral heath management on soils and landscapes**

M. Doorenbosch and J. M. van Mourik

the old surface and several sods. In addition the soil profile underneath both barrows were sampled. Results are shown in Fig. 8, 9, and 10. The pollen spectra from the old surfaces and sods consist mainly of herbal pollen, dominated by heather (*Calluna vulgaris*) and less, but still in considerable amounts by grasses. This indicates that the Echoput barrows were both built in an open space dominated by heather. The open spaces were surrounded by forest vegetation, namely oak, hazel and alder. The open heath areas were probably much larger than at the older barrows described in the first two examples. At the Echoput the ADF was around 200–300 m. The open spaces were not recently created before the barrows were built. The heath vegetation had already had some time to establish and to develop and the open place must have existed some time before the barrows were built. The pollen diagrams shown in Fig. 8b and c confirm this. These diagrams show the vegetation development from a certain period prior to the barrow building. Although the soil profile have not been dated, it is clear that heath was already present some time before the barrows were constructed. The diagrams show a decrease of the surrounding forest and an increase of the heath vegetation and at the time the burial mounds were built vegetation was dominated by heather.

## 3.4   Bedafse Bergen

The Bedafse Bergen is a biogenic land dune ridge western of Rakt, a historical complex of a hamlet and arable fields, surrounded by coppice. West of the hamlet cattle herder heathland was present. Eventually a plaggic Anthrosol developed on the arable fields (Fig. 11, phase A). After AD 1000 these heaths were subjected to severe degradation. The degradation of the heaths was caused by complete deforestation during the 11th until 13th century (van Mourik et al., 2013). This deforestation triggered the first regional extension of (older) sand drifting; aeolian eroded sand was transported by the southwest winds from the heaths to Rakt. The coppice hedge around the hamlet served as protecting screen and initiated the building of a ridge of inland dunes, the Bedafse Bergen (Fig. 11, phase B). After that the introduction of the deep stable economy in the 18th century (Vera, 2011) initiated the second extension of (younger)

sand drifting (Fig. 11, phase C). At the east side of the ridge, the western edge of the plaggic Anthrosol was buried by (younger) driftsand, at the west side of the ridge we could excavate the (by older drift sand) buried Podzol in coversand. Both profiles were sampled for pollen analysis and optically stimulated Luminescence (OSL) dating. The

Ah of the buried Podzol in profile Bedafse Bergen was also sampled for radiocarbon ($^{14}$C) dating. The of development landscape around the Bedafse Bergen is presented in Fig. 11, the sampled profiles in Fig. 12. Pollen diagram Rakt in Fig. 13, pollen diagram Bedafse Bergen in Fig. 14.During the transformation of heaths into new arable land and forests in the 20th century, the majority of the soil archives have been destroyed.

For this reason, the soil archives of the Bedafse Bergen have a high scientific value.

The position of the sampled profile is indicated in Fig. 9c. The oldest formation is coversand. The post-sedimentary pollen spectra of the 3Ap in coversand demonstrate that arable land was created on *Calluna* heath. Radiocarbon dates of similar profiles indicate a start of sedentary agriculture around 1000 BC (van Mourik, 2012, 2013). The

OSL dating (L1) of the 3Ap reflect ploughing of the agricultural soil until around AD 100 (ploughing resulted in bleaching of quartz grains, originally part of the coversand deposit). Until ≈ AD 1600 the plaggic deposition rate was relatively low (2Aan2). After AD 1600 the OSL datings points to an acceleration of the plaggic deposition (2Aan1), related to an increasing content of mineral grains of the plaggic manure. It is known

that in the course of the 18th century farmers used, in addition to straw, *Calluna* heath sods as stable filling (van Mourik et al., 2016). The difference in acceleration rate is also reflected by the pollen density curve. The plaggic Anthrosol was buried by (younger) driftsand around AD 1800.

The currently buried Podzols of the profiles Bedafse Bergen and Rakt developed

originally at the same stratigraphic level in coversand. The position of the sampled profiles is indicated in Fig. 9c. The Podzol Bedafse Bergen is currently buried below 12 m high land dune, only the basic layers have been sampled for pollen analysis and dating of the basis of the older driftsand deposits. The (sin-sedimentary) pollen spectra of the horizons of the buried Podzol contain high percentages of Ericaceae, pointing to

**SOILD**

doi:10.5194/soil-2015-83

**The impact of ancestral heath management on soils and landscapes**

M. Doorenbosch and J. M. van Mourik

Discussion Paper | Discussion Paper | Discussion Paper | Discussion Paper

the presence of heath. The relatively high percentages of *Quercus* and *Corylus* refer to deciduous forest in the surroundings. The pollen spectra of the driftsand layers show lower percentages of Ericaceae and higher percentages of Poaceae, indicating heath degradation. The decrease of *Quercus* refers to deforestation. The oldest driftsand layers have been deposited between ≈ AD 1300 and ≈ 1500. This correlates to the documental archived period of deforestation (Vera, 2011). The radiocarbon age of the humic acids, extracted from the 3Ah is ≈ AD 725. It is known that radiocarbon ages of humic horizons of palaeosols overestimate the real ages (van Mourik et al., 2010).

## 4   Discussion

### 4.1   The barrow heath landscape and heath management

Based on the examples given above and many other palynologically investigated barrows (Doorenbosch, 2013) it can be concluded that barrows were built in open spaces that varied in size from small, with an average distance to the forest (ADF) of ≈ 50–100vm for the oldest barrows, to rather large (ADF ≈ 300 m) for the younger Iron Age barrows. These open spaces were in several cases connected to each other, forming long-stretched heath areas. In the case of the Echoput barrows it was shown that the open space with heath vegetation was already present some time before the barrows were built. For the remaining investigated barrows such a vegetation history has not been reconstructed. However, in all cases heath and herb vegetation had already developed at the barrow places. This suggests that these heath areas were already present some time *before* the barrows were built in all cases. This is also confirmed by several studies on early pre-barrow palaeosols in for example the Laarder Wasmeer area (Sevink et al., 2013) and the Schaijkse Heide (van Mourik et al., 2013). In these areas very early sand-drifting periods were recorded (respectively around 4000 and 4700 cal BC). In the pollen diagrams derived from these soils it was shown that heath was already continuously present before the sand-drifting events.

Discussion Paper | Discussion Paper | Discussion Paper | Discussion Paper | Discussion Paper |

**SOILD**

doi:10.5194/soil-2015-83

**The impact of ancestral heath management on soils and landscapes**

M. Doorenbosch and J. M. van Mourik

The open spaces covered with heath vegetation must have been kept open until the barrows were built and in most cases they were kept open long after the barrows had been built. To maintain the heath, the landscape must have been managed. When heath is not managed, other plant species will replace the heath vegetation.
Management activities like grazing, sod cutting and burning could have been carried out to maintain the heath vegetation. Sod cutting must have taken place for the purpose of the barrow building, but the amount of sods needed to build a barrow is not sufficient to explain the size of the heath areas. Burning might have been part of the heath management activities. Indications for burning have only been found in
a few cases by the recording of charcoal that was probably not just related to the burial itself. Indications for large-scale burning of the heath areas have not been found however. Grazing indicators such as grasses (Poaceae), *Plantago lanceolata*, *Asteraceae liguliflorae*, *Succisa* and *Galium*-type could be an indication that the heathlands have been grazed (Hjelle, 1999). Such grazing indicators are present in all
the presented pollen spectra (Figs. 4, 6, 8), making it likely that grazing was involved in the heath management activities of prehistoric man.

Archaeozoological evidence from several excavations suggests that prehistoric farming communities kept mainly sheep and cattle (Brinkkemper and van Wijngaarden-Bakker, 2005, p. 493). In present times both sheep and cows are used to maintain
heathland areas by grazing. Young *Calluna* heath is in favour by sheep, together with grass and herb vegetation. They are not very fond of older *Calluna* plants (Elbersen et al., 2003). Cattle prefer mainly grasses, although some landraces also eat young *Calluna* plants (Lake et al., 2001, p. 31). To prevent grasses and other plant species from getting dominant in heathland areas 1 sheep per hectare and/or 1 head per 5–
6 ha is required (Elbersen et al., 2003; Siebel and Piek, 2001; Verbeek et al., 2006). The number of livestock belonging to late Neolithic farming communities has not been estimated, but for the Middle Bronze Age B has been suggested that a livestock of up to 30 animals could be kept per household (Fokkens, 2005, p. 427). For the Iron Age, estimated numbers are lower (Brinkkemper and van Wijngaarden-Bakker, 2005).

**SOILD**

doi:10.5194/soil-2015-83

**The impact of ancestral heath management on soils and landscapes**

M. Doorenbosch and J. M. van Mourik

Title Page

Abstract | Introduction

Conclusions | References

Tables | Figures

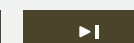 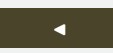 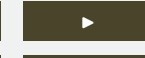 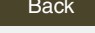 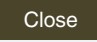

Assuming the amount of barrows was numerous, with each barrow lying in a heath area of approximately 3 ha (based on an ADF of 100 m), several households, forming heath communities, must have worked together to maintain the barrow heathland. Whether prehistoric man carried out the heath management activities ascribed above, especially
grazing, with the aim of managing the heath is not known for certain. They might have been practiced as part of their daily (agricultural) activities, while managing the heath was incidental. However, they must have realised what impact their grazing activities had on the heathland. Since overgrazing leads to the destruction of the heathland, causing possible sand drifting events, and with grazing pressure too low the heath
vegetation will be taken over by other species, prehistoric man must have adjusted their grazing activities in such a manner that the heath was preserved for thousands of years. For certain, heath was very important for prehistoric man. They might have considered the heaths as ancestral, since they did not only serve as burial placed for ancestors, they also had been used and managed by these ancestors prior to the
barrow building. As such the heaths formed part of the economic zone of the people living in the area. The ancestral heaths were very stable elements in the landscape and were kept in existence for thousands of years, forming the most important factor in structuring the barrow landscape.

## 4.2   Heath in the Roman Period

Unfortunately not much is known about heath and the maintenance of heath during the Roman Period. In the Netherlands the Romans settled mostly in the fluvial district, not in the coversand area. The practice of barrow building was no longer continued and no soil records have been investigated that could give information on the use of heathland in the period thereafter. There are some studies, however, that suggest that heathland
areas continuously existed throughout the Roman Period, indicating that some form of heath management must have taken place during that time, for example at the Echoput (see Sect. 3.3). At the Echoput heath was present at and prior to the period the barrows were built in the late Iron Age. This heath area must have been managed for some

**SOILD**

doi:10.5194/soil-2015-83

**The impact of ancestral heath management on soils and landscapes**

M. Doorenbosch and J. M. van Mourik

Discussion Paper | Discussion Paper | Discussion Paper | Discussion Paper

time to maintain. At the same site several post holes have been discovered close to the barrows. These post holes, which probably date to the Late Medieval Period, have been analysed for pollen as well (Doorenbosch 2013, chapter 8.1). It was shown that at the time the posts were placed the heath had expanded compared to when the barrows
were built. It cannot be said with certainty that the heath was present and maintained in the period in between the barrow building and the post placing, but it is likely that this is the case. The presence of heath during the Roman Period was also shown in pollen diagrams from several palaeosols, for example Venloop (Maashorst, North-Brabant, van Mourik et al., 2013) and Defensiedijk (Weert, Middle Limburg, van Mourik
et al., 2010). In addition, the presence of heath during the Roman period is mentioned in several other studies (Bakels, 2014; Kooijstra, 2008; Kooijstra and Groot, 2015). With the continuous presence of heath it is most probable that the local population continued with extensive grazing management on the heaths until the early Middle Ages.

## 4.3 Heath management since AD 500: plaggic agriculture, sand drifting,
reclamation and restoration

In the Early Middle Ages people learned to collect ectorganic materials as stable fillings to produce manure, litter from the last remains of the deciduous forests and grass sods from the brook valleys (van Mourik and Jansen, 2016). Serious degradation of heaths during this time is not recorded in the soil archives. Heath management was most
20 probably restricted to burning and mowing of older *Calluna* shrubs (Burny, 1999).

During the 11th until 13th century landowners cut the last forests, because the prices of wood were going up (Vera, 2011). These deforestations resulted in the first regional extension of sand drifting (van Mourik and Jansen, 2016). To acquire fuel, farmers dug sods of the ectorganic horizon of the moist heaths (Burny, 1999), but after the
25 remove of the ectorganic sods, the moist heaths will recover in two until four years and sand drifting was not an issue. From archived documents it is known that the farmers protected the dry heaths carefully against sand drifting (Vera, 2011). But in the course of the 18th century the combination of population growth and increase of food

**SOILD**

doi:10.5194/soil-2015-83

**The impact of ancestral heath management on soils and landscapes**

M. Doorenbosch and J. M. van Mourik

Title Page

Abstract | Introduction

Conclusions | References

Tables | Figures

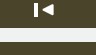 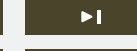

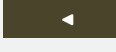 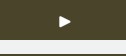

demand resulted in the extension of arable fields and the increase of the production of stable manure. Lack of stable fillings was compensated by the use of humic sods from the dry heaths (Vera, 2011; Burney, 1999). Mowing and burning were sustainable management rules, but sod digging caused degradation and initiated the second wave of sand drifting (van Mourik et al., 2012, 2013).

At the end of the 19th century, the plaggic agriculture came to an end due to the combination of the fall of prices of wool and the introduction of chemical fertilizers and urban compost. The heaths lost their economic value and the government started with the reclamation of the healthy heaths into new arable fields (after AD 1950 the production fields of mays for the extension of bio industry). The damaged heath (due to sand drifting) were turned into forests, mainly Scotch pine plantations. Figure 2 shows how the heath surface diminished from $\approx 600$ kha in 1850 to $\approx 30$ kha in 1990. Since 1980 the government started the development of the national ecological master structure to improve the biodiversity on national and regional scale. The program Natura 2000 included the preservation of the remaining heaths and restoration of lost heaths. As a result the heath surface had increased from 30 to 35 kha in 2008. Most of these heaths are now parts of protected nature reserves, in which also the (last) valuable soil archives as barrows, buried podzols and plaggic horizons will be conserved as elements of the geological and cultural heritage.

Due to the present increased nitrogen concentrations in rain and ground water, heaths cannot survive without management measures to prevent an accelerated succession to brushwood and forest. Applied measures are, just as in the 19th century, intensive grazing, mowing, burning and sod digging (Smits and Noordijk, 2013).

## 5 Conclusions

- The invasion of heaths into the coversand landscape of the Netherlands is associated with forest degradation, at first by natural causes, after that by anthropogenic deforestations.

Discussion Paper | Discussion Paper | Discussion Paper | Discussion Paper |

**SOILD**

doi:10.5194/soil-2015-83

**The impact of ancestral heath management on soils and landscapes**

M. Doorenbosch and J. M. van Mourik

- People created and maintained the barrow landscape on ancestral heaths from the Late Neolithic until the late Iron Age.

- For these people, heath had not only economic but also cultural values.

- During and after the Roman Time, people continued with heath management, mainly by cattle grazing; the heaths maintained their economic, but lost the cultural value.

- Introduction of the plaggic agriculture system around AD 500 resulted in soil acidification of the heaths and the development of plaggic horizons on arable fields.

- Deforestations during the 11th until 13th century caused extension of sand drifting and the introduction of the deep stable economy in the 18th century resulted in extensions of the driftsand landscape and heath degradation.

- At the end of the 19th century the heaths lost their economical functions and were transformed into modern production fields.

- The loss of the heath area was accompanied by the loss of soil archives.

- Preservation of the remaining soil archives is crucial for the geological heritage and future research.

Discussion Paper | Discussion Paper | Discussion Paper | Discussion Paper | Discussion Paper |

**SOILD**

doi:10.5194/soil-2015-83

**The impact of ancestral heath management on soils and landscapes**

M. Doorenbosch and J. M. van Mourik

*Acknowledgements.* The study of the barrows resulted from the project "Ancestral Mounds at the Leiden University, funded by NOW, Netherlands Organization for Scientific Research". We are grateful to Jakob Wallinga (NCL, Wageningen University) for the performance of the OSL datings of the profiles Rakt and Bedafse Bergen.

For the [14]C datings of the barrows of the Echoput and the buried Podzol of profile Bedafse Bergen we thank Hans van der Plicht (CIO, University of Groningen). We thank Jan van Arkel (IBED, University of Amsterdam) for the production of some of the digital illustrations.

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

Doorenbosch, M.: An environmental history of the Echoput barrows, Chapter 5, in: Iron Age
Echoes: Prehistoric land management and the creation of a funerary landscape – the "twin

# SOILD

doi:10.5194/soil-2015-83

**The impact of ancestral heath management on soils and landscapes**

M. Doorenbosch and
J. M. van Mourik

Title Page

| Abstract | Introduction |
| Conclusions | References |
| Tables | Figures |

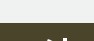

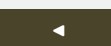 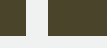

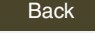 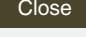

barrows" at the Echoput in Apeldoorn, edited by: Fontijn, D. R., Bourgeois, Q. P. J., and Louwen, A., Sidestone Press, Leiden, 111–128, 2011.

Doorenbosch, M.: Ancestral Heaths. Reconstructing the barrow landscape in the central and southern Netherlands, PhD thesis, University of Leiden, Leiden, 2013.

Elbersen, B. S., Kuiters, A. T., Meulenkamp, W. J. H., and Slim, P. A.: Schaapskuddes in het natuurbeheer. Alterra, Research Instituut voor de Groene Ruimte, Wageningen, 2003.

Fokkens, H.: Mixed farming societies: synthesis, in: Prehistory of the Netherlands, edited by: Louwe Kooijmans, L. P., van den Broeke, P. W., Fokkens, H., and van Gijn, A. L., Amsterdam University Press, Amsterdam, 463–474, 2005.

Groenman-van Waateringe, W.: Palynologisch onderzoek van het urnenveld te Weert, KNAG/UvA, Amsterdam, Nederlandse Geografische Studies, 74, 59–137, 1988.

Hjelle, K. L.: Modern pollen assemblages from mown and grazed vegetation types in western Norway, Rev. Palaeobot. Palyno., 107, 55–81, 1999.

Kooijstra, L. I.: Vegetation history and agriculture in the cover-sand area west of Breda (province

of Noord-Brabant, The Netherlands), Veg. Hist. Archaeobot., 17, 113–125, 2008.

Kooijstra, L. I. and Groot, M.: The agricultural basis of the Hoogeloon villa and the wider region, in: The Roman Villa of Hoogeloon and the Archaeology of the Periphery, edited by: Roymans, N., Derks, T., and Hiddink, H., Amsterdam University Press, Amsterdam, 141–162, 2015.

Lake, S., Bullock, J. M., and Hartley, S.: Impacts of livestock grazing on lowland heathland in the

UK, NERC Centre for Ecology and Hydrology and Sussex University, Peterborough, 2001.

LNV: Handboek Natuurdoeltypen, Rapport Expertisecentrum LNV 2001/20, Wageningen, 2001.

LNV: Natura 2000 doelendocument, Rapport LNV 2006/1.1, Den Haag, 2006.

Moore, P. D., Webb, J. A., and Collinson, M. E.: Pollen analysis, Blackwell Scientific Publication,

Oxford, 1991.

Sevink, J., Koster, E. A., van Geel, B., and Wallinga, J.: Drift sands, lakes and soils: the multiphase Holocene history of the Laarder Wasmeren area near Hilversum, The Netherlands, Neth. J. Geosci., 92, 243–266, 2013.

Siebel, H. and Piek, H.: Veranderde inzichten over begrazing bij natuurbeheerders, Vakblad

Natuurbeheer, 4, 45–49, 2001.

Smits, J. and Noordijk, J.: Heidebeheer, moderne methoden in een eeuwenoud landschap, KNNV uitgeverij, 2013.

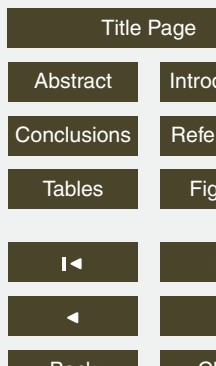

**SOILD**

doi:10.5194/soil-2015-83

**The impact of ancestral heath management on soils and landscapes**

M. Doorenbosch and J. M. van Mourik

Spek, T.: Het Drentse esdorpenlandschap, Een historisch geografische studie, Matrijs, Utrecht, 2, part VI: Plaggenbodems an Plaggenlandbouw, 725–967, 2004.

van der Linde, C. and Fontijn, D. R.: Mound 1 – A monumental Iron Age barrow, Chapter 2, in: Iron Age Echoes: Prehistoric land management and the creation of a funerary landscape – the "twin barrows" at the Echoput in Apeldoorn, edited by: Fontijn, D. R., Bourgeois, Q. P. J., and Louwen, A., Sidestone Press, Leiden, 33–64, 2011.

van Geel, B., Coope, G. R., and van der Hammen, T.: Palaeoecology and stratigraphy of the Lateglacial type section at Usselo (The Netherlands), Rev. Palaeobot. Palyno., 60, 25–129, 1989.

van Giffen, A. E.: Een Meerperioden-heuvel, Tumulus I te Bennekom, gem. Ede, Bijdragen en Mededelingen van de Vereniging Gelre, LIV, 1–21, 1954.

van Mourik, J. M.: The use of micromorphology in soil pollen analysis The interpretation of the pollen content of slope deposits in Galicia, Spain, Catena, 35, 239–257, 1999.

van Mourik, J. M.: Pollen and spores, preservation in ecological settings, in: Palaeobiology, edited by: Briggs, D. E. G. and Crowther, P. R., Blackwell Science, II, 315–318, 2001.

van Mourik, J. M. and Slotboom, R. T.: The expression of the tripartition of the Allerød chronozone in the lithofacies of Late Glacial polycyclic profiles in Belgium and the Netherlands, Mededelingen Rijks Geologische Dienst, 52, 441–450, 1995.

van Mourik, J. M., Nierop, K. G. J., and Vandenberghe, D. A. G.: Radiocarbon and optically stimulated luminescence dating based chronology of a polycyclic driftsand sequence at Weerterbergen (SE Netherlands), Catena, 80, 170–181, 2010.

van Mourik, J. M., Slotboom, R. T., and Wallinga, J.: Chronology of plaggic deposits; palynology, radiocarbon and optically stimulated luminescence dating of the Posteles (NE-Netherlands), Catena, 84, 54–60, 2011.

van Mourik, J. M., Seijmonsbergen, A. C., and Jansen, B.: Geochronology of Soils and Landforms in Cultural Landscapes on Aeolian Sandy Substrates, Based on Radiocarbon and Optically Stimulated Luminescence Dating (Weert, SE-Netherlands), Radiometric Dating, 75–114, 2012.

van Mourik, J. M., Seijmonsbergen, A. C., Slotboom, R. T., and Wallinga, J.: The impact of human land use on soils and landforms in cultural landscapes on aeolian sandy substrates (Maashorst, SE Netherlands), Quatern. Int., 265, 74–89, 2013.

**SOILD**

doi:10.5194/soil-2015-83

**The impact of ancestral heath management on soils and landscapes**

M. Doorenbosch and J. M. van Mourik

van Mourik, J. M., Wagner, T., de Boer, W. G., and Jansen, B.: The added value of biomarker analysis to the genesis of Plaggic Anthrosols; the identification of stable fillings used for the production of plaggic manure, SOIL Discuss., in press, 2016.

Vera, H.: ... dat men het goed van den ongeboornen niet mag verkoopen, Gemene gronden in de Meierij van Den Bosch tussen hertog en hertgang 1000–2000. Uitgeverij BOXpress, Oisterwijk, Netherlands (with English summary), 2011.

Verbeek, P. J. M., M. de Graaf, and Scherpenisse, M. C.: Verkennende studie naar de effecten van drukbegrazing met schapen in droge heide: effectgerichte maatregel tegen vermesting in droge heide, Directie Kennis, Ministerie van Landbouw, Natuur en Voedselkwaliteit, Ede, 51 pp., 2006.

Waterbolk, H. T.: De praehistorische mens en zijn milieu, Een palynologisch onderzoek naar de menselijke invloed op de plantengroei van de diluviale gronden in Nederland, University of Groningen, Netherlands, 1954.

Weeda, E. J., Westra, R., Westra, Ch., and Westra, T.: Nederlandse Oecologische Flora, deel 3, Heidefamilie, Uitgave IVN in samenwerking met VARA en VEWIN, 30–53, 1988.

**SOILD**

doi:10.5194/soil-2015-83

**The impact of ancestral heath management on soils and landscapes**

M. Doorenbosch and J. M. van Mourik

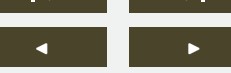

**SOILD**

doi:10.5194/soil-2015-83

**The impact of ancestral heath management on soils and landscapes**

M. Doorenbosch and J. M. van Mourik

**Table 1.** OSL datings of profile Rakt (Fig. 10, left).

| Soil horizon | OSL sample ring | OSL ages |
|---|---|---|
| C top | L7 | 1812± AD 9 |
| C bottom | L6 | 1808± AD 8 |
| 2Aan1 bottom | L5 | 1781± AD 9 |
| 2Aan2 top | L4 | 1685± AD 14 |
| 2Aan2 middle | L3 | 1593± AD 17 |
| 2Aan2 bottom | L2 | 1417± AD 37 |
| 3Ap | L1 | 0069± AD 89 |

**SOILD**

doi:10.5194/soil-2015-83

**The impact of ancestral heath management on soils and landscapes**

M. Doorenbosch and J. M. van Mourik

**Table 2.** Absolute dating of profile Bedafse Bergen (Fig. 10, right).

| Horizon | OSL ring/[14]C box | OSL dating | [14]C humic acids |
|---|---|---|---|
| 2C2,3 | L3 | 1473± AD 40 | |
| 2C4,5 | L1 | 1425± AD 40 | |
| 2C6,7 | L1 | 1365± AD 40 | |
| 3Ah | RC | | 725± AD 39 |

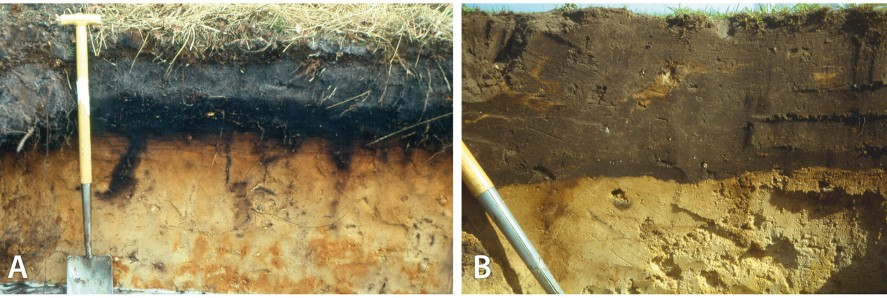

**Figure 1.** Destruction of soil archives on the Maashorst (North Brabant) during the transition of (former) heaths into modern arable fields. **(a)** Undisturbed gleyic Podzol, **(b)** deeply ploughed Podzol.

Discussion Paper | Discussion Paper | Discussion Paper | Discussion Paper | Discussion Paper |

**SOILD**

doi:10.5194/soil-2015-83

**The impact of ancestral heath management on soils and landscapes**

M. Doorenbosch and J. M. van Mourik

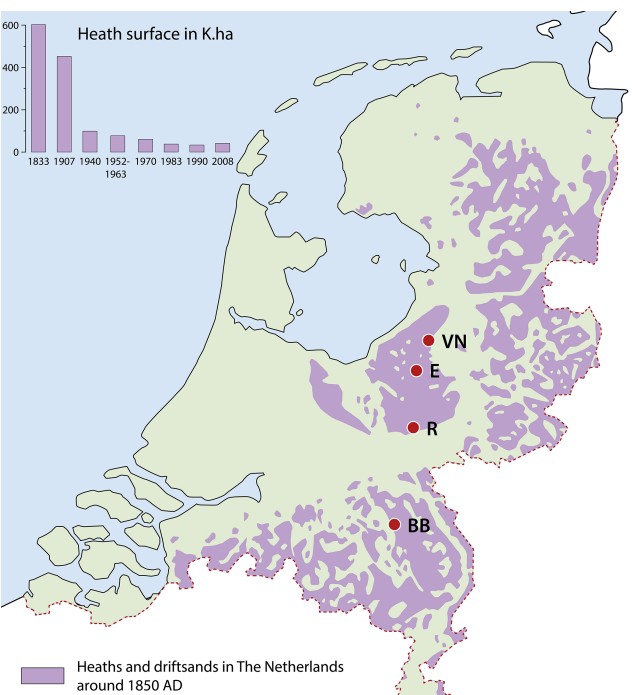

**Figure 2.** Heath area in the Netherlands around AD 1850 and the diminution of heaths between 1833 and 2008. The locations of the four research sites are indicated: VN: Vaassen-Niersen; R: Renkum stream valley; E: Echoput; BB: Bedafse Bergen.

Discussion Paper | Discussion Paper | Discussion Paper | Discussion Paper

**SOILD**

doi:10.5194/soil-2015-83

**The impact of ancestral heath management on soils and landscapes**

M. Doorenbosch and J. M. van Mourik



# SOILD

doi:10.5194/soil-2015-83

**The impact of ancestral heath management on soils and landscapes**

M. Doorenbosch and
J. M. van Mourik

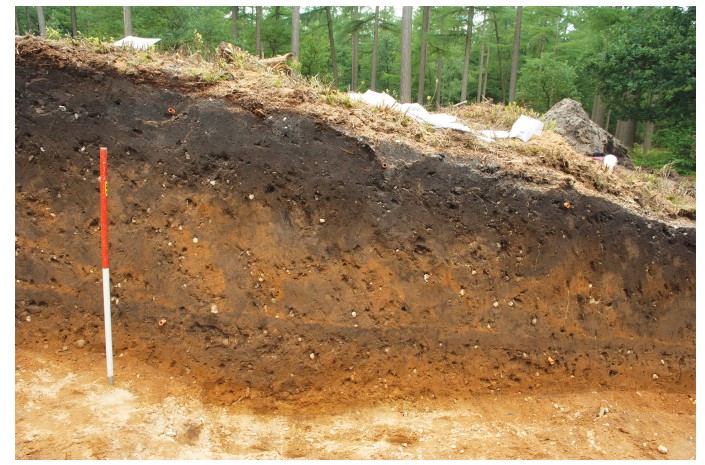

**Figure 3.** Soil record of a barrow; showing a palaeosol buried by a barrow, built with sods, dug in heathlands in the surroundings. Photograph by Q. Bourgeois.

**SOILD**

doi:10.5194/soil-2015-83

**The impact of ancestral heath management on soils and landscapes**

M. Doorenbosch and J. M. van Mourik

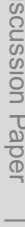

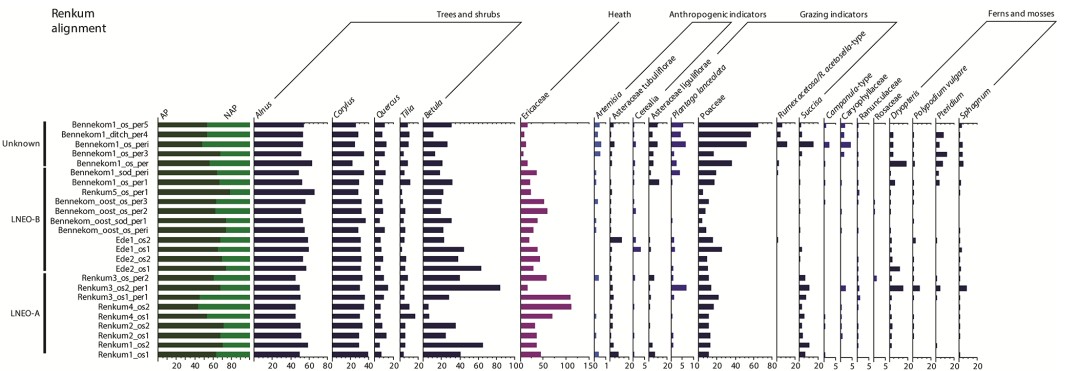

**Figure 4.** Pollen spectra from the samples taken from the barrows of the Renkum alignment.

Discussion Paper | Discussion Paper | Discussion Paper | Discussion Paper | Discussion Paper |

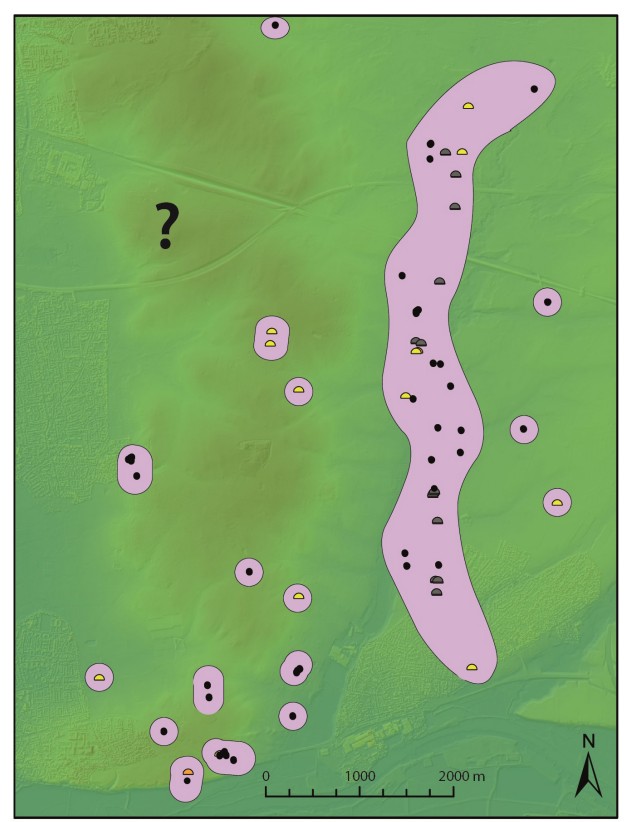

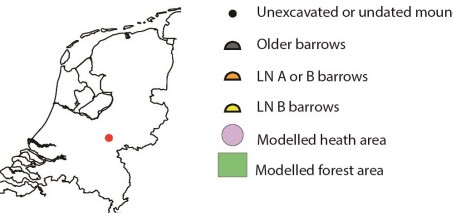

- • Unexcavated or undated mounds
- ⬖ Older barrows
- ⬖ LN A or B barrows
- ⬖ LN B barrows
- ⬤ Modelled heath area
- ⬛ Modelled forest area

**SOILD**

doi:10.5194/soil-2015-83

**The impact of ancestral heath management on soils and landscapes**

M. Doorenbosch and J. M. van Mourik

Discussion Paper | Discussion Paper | Discussion Paper | Discussion Paper | Discussion Paper

**Figure 5.** Barrow alignments of Renkum, situated in a (hypothetical) long-stretched heath area surrounded by forest. The vegetation reconstruction is based on palynological data from barrows. An exact reconstruction of the forest area is therefore not possible (indicated by the question mark), since barrows are not present in those areas. The figure is based on the digital elevation model of the AHN (copyright www.ahn.nl). Figure after Doorenbosch (2013, Fig. 13.2c).

SOILD

doi:10.5194/soil-2015-83

The impact of ancestral heath management on soils and landscapes

M. Doorenbosch and J. M. van Mourik

Discussion Paper | Discussion Paper | Discussion Paper | Discussion Paper | Discussion Paper |

**SOILD**

doi:10.5194/soil-2015-83

**The impact of ancestral heath management on soils and landscapes**

M. Doorenbosch and J. M. van Mourik

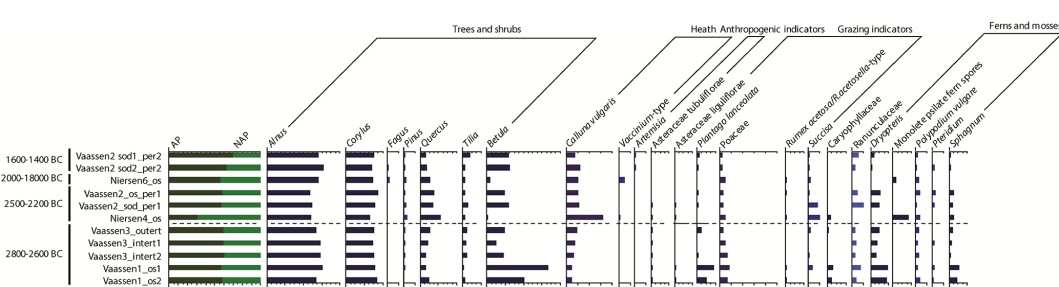

**Figure 6.** Pollen spectra from samples taken from the barrows at Vaassen, the barrows at Niersen barrows and the Celtic Field at Vaassen.



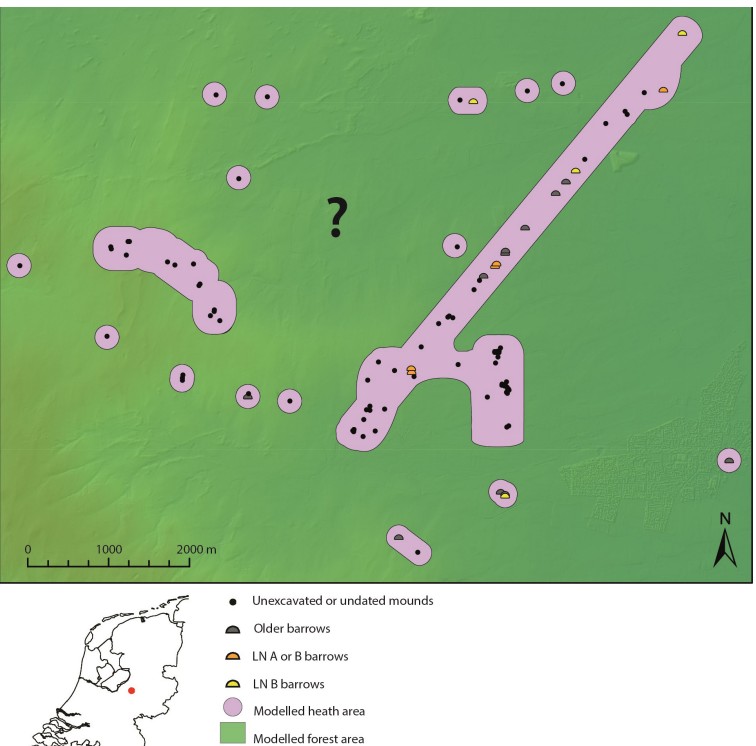

**Figure 7.** Barrow alignments of Vaassen-Niersen, situated in a (hypothetical) long-stretched heath area surrounded by forest. The vegetation reconstruction is based on palynological data from barrows. An exact reconstruction of the forest area is therefore not possible (indicated by the question mark), since barrows are not present in those areas. The figure is based on the digital elevation model of the AHN (copyright www.ahn.nl). Figure after Doorenbosch (2013, Fig. 13.1c).

Discussion Paper | Discussion Paper | Discussion Paper | Discussion Paper

**SOILD**

doi:10.5194/soil-2015-83

**The impact of ancestral heath management on soils and landscapes**

M. Doorenbosch and J. M. van Mourik

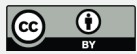

Discussion Paper | Discussion Paper | Discussion Paper | Discussion Paper | Discussion Paper

**SOILD**

doi:10.5194/soil-2015-83

**The impact of ancestral heath management on soils and landscapes**

M. Doorenbosch and J. M. van Mourik

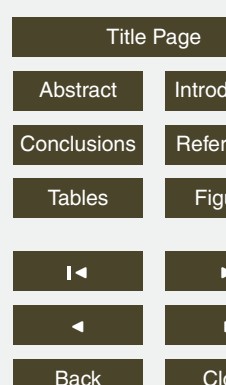

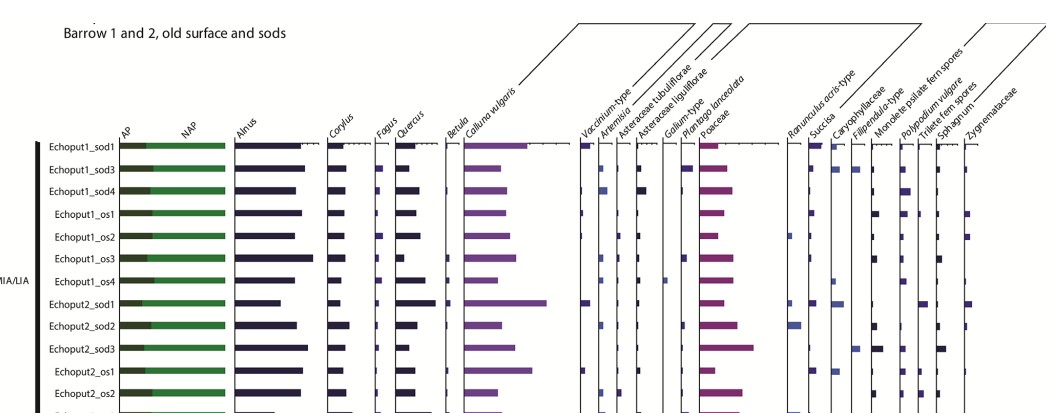

**Figure 8.** Pollen spectra from the sod and old surface samples taken from Echoput barrow 1 and 2.



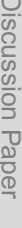

**SOILD**

doi:10.5194/soil-2015-83

**The impact of ancestral heath management on soils and landscapes**

M. Doorenbosch and J. M. van Mourik

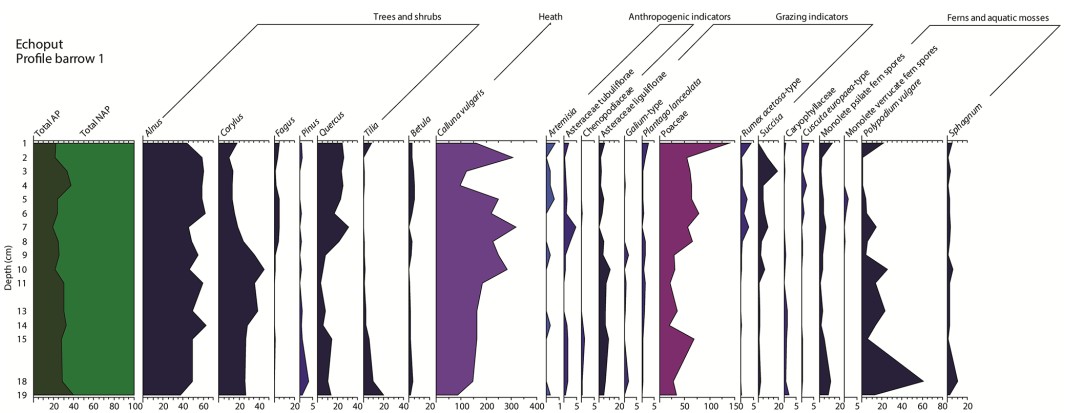

**Figure 9.** Echo **(b–c)**. Pollen diagrams derived from the series of samples taken from underneath Echoput barrow 1.

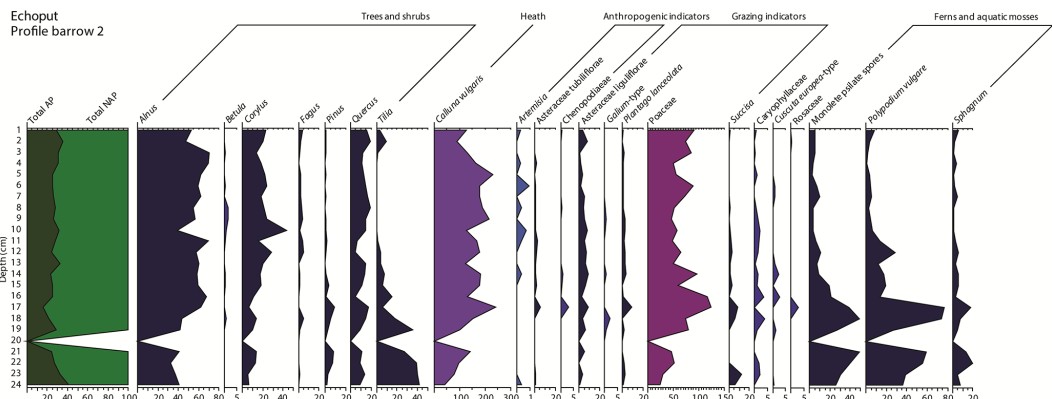

**Figure 10.** Echo **(b–c)**. Pollen diagrams derived from the series of samples taken from underneath Echoput barrow 2.

Discussion Paper | Discussion Paper | Discussion Paper | Discussion Paper | Discussion Paper |

**SOILD**

doi:10.5194/soil-2015-83

**The impact of ancestral heath management on soils and landscapes**

M. Doorenbosch and J. M. van Mourik

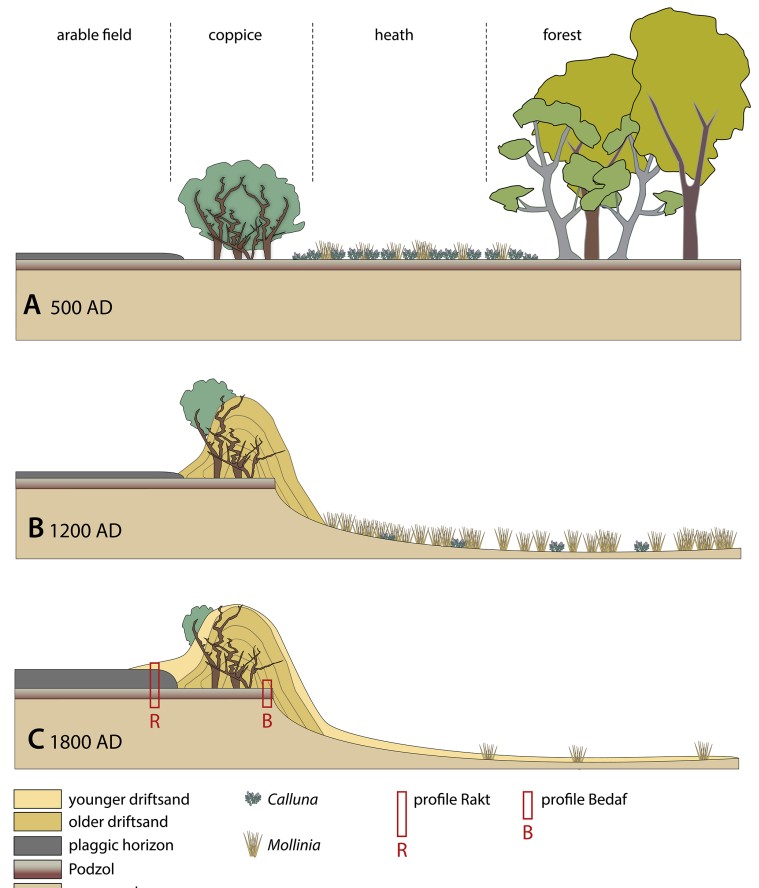

**Figure 11.** Sequence of development phases of the Bedafse Bergen and its soil archives.

# SOILD

doi:10.5194/soil-2015-83

**The impact of ancestral heath management on soils and landscapes**

M. Doorenbosch and J. M. van Mourik

Discussion Paper | Discussion Paper | Discussion Paper | Discussion Paper

SOILD

doi:10.5194/soil-2015-83

**The impact of ancestral heath management on soils and landscapes**

M. Doorenbosch and J. M. van Mourik

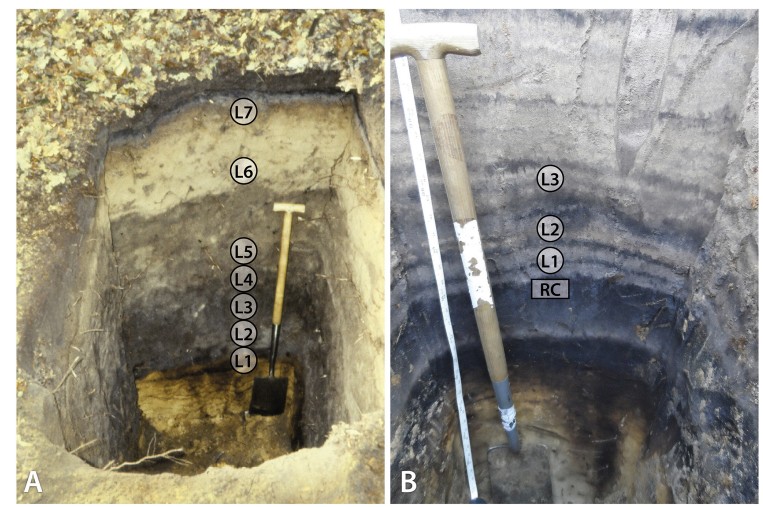

**Figure 12.** The soil archives of the Bedafse Bergen (A: Rakt; B: Bedafse Bergen). The sample locations for OSL are indicated with rings, for $^{14}$C with a cube.

Discussion Paper | Discussion Paper | Discussion Paper | Discussion Paper

**SOILD**

doi:10.5194/soil-2015-83

**The impact of ancestral heath management on soils and landscapes**

M. Doorenbosch and J. M. van Mourik

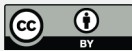

POLLEN DIAGRAM RAKT

**Figure 13.** Pollen diagram Rakt.

SOILD

doi:10.5194/soil-2015-83

**The impact of ancestral heath management on soils and landscapes**

M. Doorenbosch and J. M. van Mourik

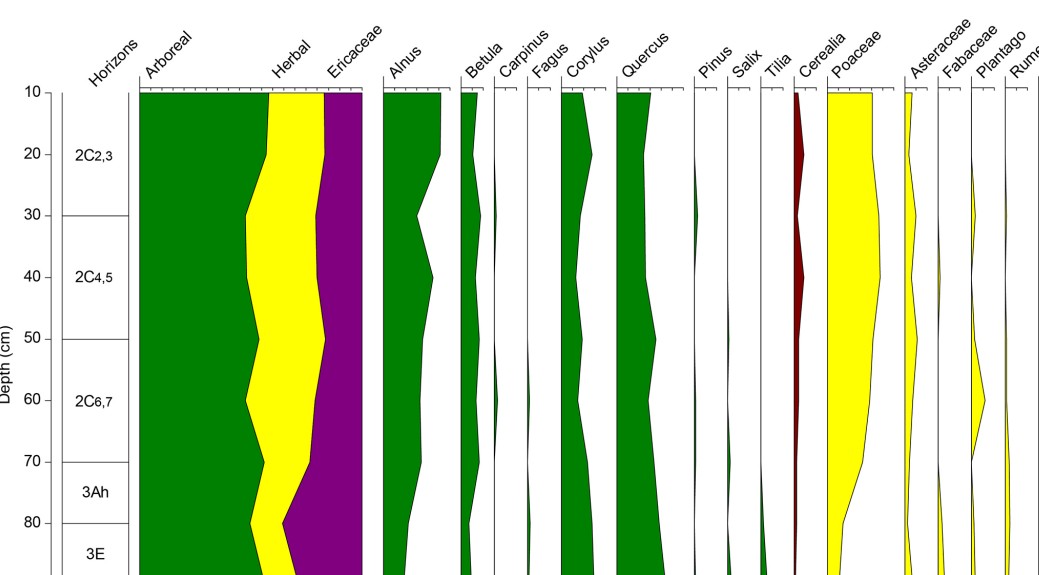

**Figure 14.** Pollen diagram Bedafse Bergen.

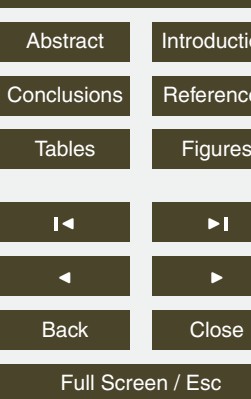