# Peer review of "Published: 18 January 2016"

_SOIL, 2015_

## Referee Comment (RC1) · S.J. Kluiving (Referee) · 15 Apr 2016

P5 L 18 typo Podzol

P7 L1-8; Make clear what data are newly acquired an processed in this study, and what data are (re-)interpreted from previous publications, e.g. Bedafse Bergen.

P11 L3: could excavate the   Podzol in coversand buried by older drift sand.

P11 L3-6: Both profiles were sampled for pollen analysis and optically stimulated Luminescence (OSL) dating. The Ah $_5$ of the buried Podzol in profile Bedafse Bergen was also sampled for radiocarbon ($_{14}$C) dating.>> give proper referencing to these 14C analyses since they are apparently not performed in this study ( see Material and Methods).

P11 L8 : Fig. 14.During  / add space

P11 L11; The position of the sampled profile is indicated in Fig. 9c. Give proper referencing here!

P11 L14-17 ; The
$_{15}$OSL dating (L1) of the 3Ap reflect ploughing of the agricultural soil until around AD100 (ploughing resulted in bleaching of quartz grains, originally part of the coversand deposit). >> data from this study? Is not referenced... If it is the first time these data (C14 and OSL)  are published, the chronological methods should be described under Methods and Materials

P12, L6-7 The radiocarbon age of the
humic acids, extracted from the 3Ah is _AD725.  Same comment as above, absolute ages have not been acquired in the framework of this study?

P14 L21-22: In the Netherlands the Romans settled mostly in the fluvial district, not in the coversand area. Hard to believe, because of the villa landscape dissertation by Karen Jeneson as well as publications by Roymans, e.g.

Roymans, N./T. Derks/H.A. Hiddink, 2015: *The Roman villa of Hoogeloon and the archaeology of the periphery*, Amsterdam (Amsterdam University Press).

Roymans, N./T. Derks (eds), 2011: *Villa landscapes in the Roman North. Economy, culture and lifestyles*, Amsterdam (Amsterdam University Press).

---

## Referee Comment (RC2) · M. Gocke (Referee) · 4 May 2016

This is a really nice and well done study with profound and carefully consulted background knowledge. I strongly recommend its publication in the SOIL special issue "Soil as a Record of the Past", as it highlights the benefits of preserving valuable soil archives in the Netherlands.

Most of my comments listed below are suggestions for minor corrections:

[Figure]

GENERALLY

- It would be nice if the term barrow and its meaning could be explained in one additional sentence. It was shortly mentioned (indirectly) in section 2.1 that these are burial mounds, but this should be explained in a more prominent way, as these are one of the main objects of research of the study.

- Please be careful to use either British or American English.

ABSTRACT

- Page 2, lines 8f: "in addition on THAT, the forest degradation...".

INTRODUCTION

- Page 3, line 10: Probably, the authors meant "cosmopolitan ligneous plant species"?

- Page 4, line 11: natural instead of naturel.

- Page 4, lines 20ff: "and soil archives were SEVERELY damaged or even destroyed...". Further, I would not use the term "heath surface" but maybe rather "heath area".

- Page 4, line 27: "...sustainably protected".

- Page 4, lines 28-29: Maybe, the authors could state here a bit more in detail, which properties (physical? chemical?) of soil and landscape they mean, that were affected by heath management.

- Page 5, line 6: "... during this period".

- Generally, it would be nice if the Introduction would terminate with the aims of the study. Of course, the last paragraph of this chapter is also part of the Introduction, but maybe it could be a bit rephrased so that it appears more "connected" to the before mentioned aim.

MATERIALS AND METHODS

- Page 5, line 18: Podzols.

- Page 5, lines 18ff: Please write ". . . with the pre-barrow land surface on top. . .".

- Page 6, line 27: This is the first place where the word biomarker appears. Biomarkers were not the object of this study and they are not further mentioned in the manuscript, therefore this is confusing. Moreover, if biomarkers are mentioned, it would be interesting to read which biomarkers are meant, e.g. alkanes, suberin, sterols?

- Page 7, lines 1 and 7: Maybe, the authors could explain in 1-2 sentences the mentioned methods despite citing previous studies, so that the reader gets a rough idea what has been done with the soil material.

RESULTS

- Page 7, lines 15ff: As already mentioned by the editor, Mr. Kluiving, here the authors do not clearly distinguish between their own results and published work.

- Page 7, lines 18f: Please mention, which fraction of the barrow material was dated.

- Page 7, line 20: Maybe the authors should state that the pollen spectra in Fig. 4 is a combination of several barrows.

- Page 8, line 26: Please rephrase.

- Page 9, line 2: From here onwards, there seems to be some chaos with the figure numbers.

- Page 9, lines 21ff: Is it common use to write these terms? Maybe this could be difficult to understand for non-archeologists.

- Page 10, lines 10f: What do you mean with "some time"? Years, decades, centuries?

- Page 11, line 6: "The development of landscape".

- Page 11, line 13: Again, please mention which fraction of the soil material was dated.

- Page 11, line 28: "syn-sedimentary"

- Generally, section 3.4 mainly discusses previous studies from Bedafse Bergen and does not show only own results. This does not reduce the value of this study, but the section or parts of it might be re-arranged / shifted to the Discussion chapter.

DISCUSSION

- Page 16, lines 10f: Please rephrase "The heath damaged by sand drifting was turned into. . .".

FIGURES AND TABLES

- Tables 1 and 2 are not referred to throughout the text.

- Figure 11: Very nice illustration of the chronological context at Bedafse Bergen.

---

## Author Comment (AC1) · 14 May 2016

We thank the reviewers for their carefully and critical comments. We will use all their suggestions to improve the quality of this paper. We will also use the comments of Jan Sevink and Bas van Geel (reviewers comment, send to the authors and not posted in the SOILD discussion menu. One important advize was: make clear which sources were used to create an overview of heath management studies and which case are used as new material. So we are sure that we can improve the quality of our paper.

[Figure]

Marieke Doorenbosch, Jan van Mourik.

---

## Author Response (AR1)

**Summary of executed changes:**

In the understanding text file all the revisions are marked (yellow). We followed the critical remarks of the reviewers and also Bas van Geel and Jan Sevink (the reviewers that send their remarks to the authors and not to the Soil discussion panel). The result is an total amount of changes that exceeds a little bit "minor revision", but we are convinced that which the aid of all the review comment we could improve the quality of this paper. No changes have been executed in the illustrations.

May 25, 2016,
The authors
* * *
**The impact of ancestral heath management on soils and landscapes: a reconstruction based on palaeoecological analyses of soil records in the middle and southeastern Netherlands**

**M. Doorenbosch[1]\* and J. M. van Mourik[2]**

[1]Faculty of Archaeology, University of Leiden, Einsteinweg 2, 2333CC Leiden, the Netherlands
[2]Institute for Biodiversity and Ecosystem Dynamics, University of Amsterdam, Science Park 904, 1098 XH Amsterdam, the Netherlands
\*Corresponding author M. Doorenbosch (m.doorenbosch@arch.leidenuniv.nl)

**Abstract**

The evolution of heath lands during the Holocene has been registered in various soil records. Palaeoecological analyses of these records enable to reconstruct the changing economic and cultural management of heaths and the consequences for landscape and soils.
Heaths are characteristic components of cultural landscape mosaics on sandy soils in the Netherlands. The natural habitat of heather species was moorland. At first, natural events like forest fires and storms caused small-scale forest degradation, in addition on that, the forest degradation accelerated due to cultural activities like forest grazing, wood cutting and shifting cultivation. Heather plants invaded on degraded forest soils and heaths developed. People learned to use the heaths for economic and cultural purposes. The impact of the heath management on landscape and soils was registered in soil records of barrows, drift sand sequences and plaggic Anthrosols. Based on pollen diagrams of such records we could reconstruct that heaths were developed and used for cattle grazing before the Bronze Age. During the Late Neolithic, the Bronze Age and Iron Age, people created the barrow landscape on the ancestral heaths. After the Iron Age people probably continued with cattle grazing on the heaths and plaggic agriculture until the Early Middle Ages. Severe forest degradation by the production of charcoal for melting iron during the Iron Age till the 6th-7th century and during the 11th-13th century for the trade of wood resulted in extensive sand drifting, a threat for the valuable heaths. The introduction of the deep stable economy and heath sods digging in the course of the 18th century resulted in acceleration of the rise of plaggic horizons, severe heath degradation and again extension of sand drifting. At the end of the 19th century heath lost its economic value due to the introduction of chemical fertilizers. The heaths were transformed into "new" arable fields and forests and due to deep ploughing most soil archives were destroyed. Since AD 1980, the remaining relics of the ancestral heaths are preserved and restored in the frame of the programs to improve the regional and national geo-biodiversity. Despite the realization of many heath restoration projects during the last decades, the area of the present heaths is just a fraction of the heath areal in 1900 AD.

**Keywords**
Heath management, soil archives, barrows, plaggic Anthrosols, drift sands, Netherlands

**1 Introduction**
Heaths are characteristic biotopes in Northwestern Europe. Most of the heaths in the study area occur on Late-glacial cover sand and Holocene drift sand deposits. The current biodiversity management regards dry and moist heaths as region specific ecotopes that need to be preserved and protected (LNV, 2001, 2006). *Erica*

*tetralix* dominated the moist, *Calluna vulgaris* the dry heaths; both species grow on acid, humic soils, poor in nitrogen and phosphate (Weeda et al., 1988; IVN, 2001). Originally, the term heath was not connected to a vegetation type, but to common fields for cattle grazing on chemically poor sandy soils. The family of Ericaceae counts ≈3000 cosmopolitan ligneous plant species of which seven occur in the Netherlands. Two factors are crucial for the survival of heaths on chemical poor sandy substrates (Weeda et al., 1988).

1. Heath plants live in symbiosis with *ericoid mycorrhiza*. These organisms provide the heath plants with nitrogen that is hardly available for other plants, which live without this form of symbiosis.

2. Heath plants have specific adaptations to drought stress. Leaves will stop evaporation of water under dry or warm air conditions, but can evaporate large amounts of water under humid air conditions. These adaptations enable heath plants to enhance nutrient uptake from soil water with low nutrient concentrations. In this way heath plants are able to concentrate the sufficient amount of nutrients necessary for plant growth from soil water that contains low nutrient concentrations.

Moors on poorly drained soils formed the natural habitats for heath plants. *Erica tetralix* dominated in the lower parts of the moors, *Calluna vulgaris* on the drier rises of *Sphagnum* peat (Weeda et al., 1988). The occurrence of Ericaceae in the cover sand landscape was reported in pollen diagrams of initial Histosols, developed during the Bølling and Allerød interstadials (van Geel et al., 1989; van Mourik and Slotboom, 1995). The Holocene migration of heath species from the moors to the cover sand landscapes was initially triggered by environmental events such as storm and forest fires, but accelerated by cultural factors such as deforestation and agriculture.

The Holocene vegetation development and soil formation on cover sands started without significant human interference. At that time, the soilscape was in accordance with the geomorphological structure, xeromorphic Podzols on cover sand ridges, gleyic Podzols on cover sand planes, histic Podzols in brook valleys. During the Early Holocene, natural events as forest fires and storms caused small scale forest degradation and created habitats outside the bogs were heath could develop. It was long thought that this situation continued until the Late Neolithic, but now there are reasons to believe that the activities of pre-agricultural communities from before ≈5000 BC already had significant impact on the landscape. This is shown by Sevink et al. (2013) in the Laarder Wasmeer area. Sevink et al. have studied several palaeosols in which very early periods of sand drifting events were recorded, OSL dated to 8800-6500, 6400-5800 and 5300-4800 years BP. Pollen analyses showed that heath was dominating the vegetation already before the earliest recorded sand drifting period. Sevink et al. concluded that, although there are no indications for farming or intensive grazing in that period, the most likely explanation for the early heath dominance associated with sand drifting was the activity of the population living in the area, since they cannot be linked to climatic changes and the youngest sand drifting period in the Laarder Wasmeren area coincides with the start of the barrow building in the area, indicating increasing intensity of the use of the area by prehistoric man. This was also suggested by Willemse and Groenewoudt (2012), who considered early Neolithic sand drifting periods in the Dutch river valleys to be of human origin.

Several other comprehensive studies, based on soil archives, were performed to explore the causes of Holocene sand drifting in the areas Drenthe (Catsel, 1991), Veluwe (Koster, 1978), SE-Netherlands (van Mourik, 2012a,b). In the pollen diagrams, presented in these studies, *Calluna* plays a significant role, but hardly any attention was paid to the role of (pre)historical heath management.

From the Late Neolithic onwards, the effect of human land use on soil and landscape increased. The deciduous forest gradually degraded by some woodcutting, forest grazing and shifting cultivation (van Mourik et al., 2012a,b) and the heaths that were already present in the area could expand. Most probably, the valuable common heaths on aeolian sandy soils were sustainably managed by the community (M. de Keyzer, 2014). However, periods of severe forest degradation and sand drifting threatened the survival of the heaths. Since the Early Iron Age, forest degradation was triggered by the use of oak trees for the production of charcoal to melt iron from bog iron ore, rattle stones and plaggic horizons (Laban et al., 1988). The production of iron resulted in extensive sand drifting in the Veluwe region from the Iron Age until the 6[th]-7[th] century (Beukenkamp en Sevink, 2005). During the 11[th]-13[th] century the clear cutting of forests for the sale of wood resulted in extensive sand drifting in the SE-Netherlands (van Mourik et al., 2012; Vera, 2011). In the course of the 18[th] century deep stable economy and digging of heath sods was introduced. At one hand this accelerated the rise of plaggic horizons, at the other hand heath degraded severely and sand drifting extended again.

Nowadays, many moist and dry heath areas in the Netherlands are protected, as they form part of national and (Pan) European ecological networks (LNV, 2001, 2006; Jongman et al., 2011). In order to preserve the heath

areas, future sustainable heath management must be based on knowledge on the origin of heath biotopes and the role of heaths in historical land use systems (Smits and Noordijk, 2013).

Information about historical land management on cover sands can be unlocked from soil archives. In the course of the 20th century, the majority of the heaths were transformed into arable land and forest plantations. Soils were deeply ploughed and soil archives were severely damaged or even destroyed (Fig. 1). As a consequence the total heath area in the Netherlands decreased from 600000 to 30000 ha (Fig. 2). Remaining parts of eligible soil archives can be found as palaeosols underneath barrows, i.e. burial mounds (Doorenbosch, 2013), polycyclic sequences in drift sand landscapes (van Mourik et al., 2012a,b; Sevink et al., 2013) and plaggic Anthrosols (Spek, 2004; van Mourik et al., 2011). Some of these archives have now been included in managed nature reserves and are considered as cultural heritage.

The aim of this paper is to reconstruct the impact of ancestral heath management on the development of soils (Pozols, Arenosols and plaggic Anthrososls) and landforms (barrows, drift sand covers and dunes) based on previously published research cases. Key sites that were used for this paper include recently investigated barrow palaeosols (Doorenbosch, 2013) and a cross section through an inland dune ridge van Mourik, 2012a). These sites will be described and discussed in detail in the following sections. The soil records from the barrow landscapes cover the period between ≈3000 BC and the Roman Time. The soil archives from the cross section of the dune ridge cover the period after the Roman Time and 1900. Few information is available in the literature about heath management from the beginning of the Roman Time until the Early Middle Ages. However, the continuity of the Ericaceae in pollen diagrams of raised bogs and palaeosols, suggests the continuity of heaths during this period, probably as extensive grazing areas, without clear registration in the soil archives.

**2 Materials and methods**

**2.1 Profile selection**

Barrows, mounds underneath or in which prehistoric human buried the dead, have been built from ≈3000 BC (the late Neolithic period) until ≈100 BC (the Late Iron Age). Around 4000 barrows are known to be still present in the Netherlands, but considering the large amount of barrows that has disappeared over time there must have been thousands of burial mounds in the Netherlands, dominating the morphology of the sandy Pleistocene landscape.

Many of these burial mounds have been excavated and sampled for pollen analysis to reconstruct the barrow environment (Fig. 3). To reconstruct environmental development, pollen spectra of samples from the mounds are less valuable, but spectra of samples of the buried Podzols underneath a barrow, with the pre-barrow land surface on top, are opportune with the restriction of the regular complications of soil pollen spectra (van Mourik, 2001). Pollen grains precipitate onto the land surface and infiltrate by bioturbation into the soil profile and reach the A, E, B and even the C horizons. Soil acidification, a regular development during the Holocene in sandy substrates, caused retrogressive activity during acid soil formation. Hence, older pollen assemblages will be preserved in the lower parts of the soil (van Mourik, 1999, 2001). When the soil was buried during the construction of the barrow, the active soil processes stopped and the soil record was conserved. The pollen grains, incorporated in the palaeosol before and during the time that barrow was built are nowadays in many cases still present.

A barrow is usually constructed of sods. These sods were taken from the upper part of the soil in the surroundings and placed upside down when building the barrow. Sods contain parts of the soil record from the place where they were taken, including pollen grains. Pollen spectra of the constructing materials and the palaeosol have been used to reconstruct the morphology of the barrow landscape (Waterbolk, 1954; Groenman-van Waateringe, 1988; Bloemers, 1988; Doorenbosch, 2013). These investigations have revealed that all studied barrows were built in heath vegetation that has been kept in existence mainly by human activities for several millennia, before, during and after the barrows were constructed. The management of heath will be further specified in the following paragraphs of this article.

For this paper we used the cases Renkum stream valley, Vaassen-Niersen and Echoput, previously published in Doorenbosch (2013), to demonstrate heath management of the barrow landscape between ≈3000 and ≈200 BC. Heath management after AD 500 caused two changes in the geomorphology of the landscape. Firstly plaggic horizons developed and the surface of arable fields raised, secondly severe heath degradation resulted in sand erosion and re-sedimentation. We used the case study Bedafse Bergen (van Mourik, 2012a) to demonstrate the impact of heath management on soils and landforms between AD 500 and 1900.

**2.2 Pollen analysis**

Pollen records of palaeosols in barrow landscapes, buried Podzols and Anthrosols provide palaeoecological information on plant species, present on site and in the region during the formation of the barrows, drift sand deposits and plaggic horizons. Previous research has shown that pollen grains, infiltrated in soils and incorporated in plaggic deposits, are well preserved in the anaerobic and acid microenvironment of excremental aggregates (van Mourik, 1999, 2001).

Samples for pollen extraction were collected in 10 mL tubes in profile pits. Pollen extractions were carried out using 10% potassium hydroxide (KOH), 10% hydrochloric acid (HCl), bromoform-ethanol (specific gravity= 2.0) and acetolysis (Moore et al., 1991, p. 50). For the identification of pollen grains, the pollen keys of Moore et al. (1991, p. 83–166) and Beug (2004) were applied. The pollen scores of the barrow records are based on a tree pollen sum minus *Betula* (in the curve of total AP [=arboreal pollen] 5 *Betula* is included). The barrow pollen data used in this paper have been previously published in Doorenbosch 2013 and will be summarized in this paper with a focus on heath and heath management. Pollen scores of the archives of Bedafse Bergen were based on a total pollen sum of arboreal and non-arboreal plant species. For the estimation of the pollen concentrations from the various soil horizons of profile Rakt, the exotic marker grain method was applied (Moore et al., 1991, p. 53). Pollen data of the profile Rakt has been published in van Mourik (2012a), the pollen diagram Bedafse Bergen has not been published before.

**3 Results**

**3.1 The Renkum stream valley**

In a stream valley near Renkum (for a location map see Fig. 2) an alignment of barrows is situated with a length of at least 4.5 km. Several barrows of this alignment, archaeologically dated to the late Neolithic A (≈2900–2500 BC) and the late Neolithic B period (≈2500–2000 BC), have been excavated and sampled for pollen. Pollen samples were taken from the old surfaces underneath the barrows and/or the sods the barrows were built of. Palynological analyses were performed by Casparie and Groenman-van Waateringe (1980, p. 24–36), with the exception of Bennekom 1. Bennekom 1 was published by van Giffen (1954). Doorenbosch (2013) reinterpreted and published the data retrieved by the above-mentioned researchers in her PhD thesis.

Figure 4 shows the pollen spectra from the barrows of the alignment in the Renkum stream valley. In the Neolithic A period Ericales pollen form a considerable part of the pollen spectra. Heath pollen tends to spread mostly within a few meters from the place where the heath is growing and pollen is produced (de Kort, 2002). This implies that the considerable percentage of heath pollen indicates that all investigated barrows in the area were constructed in an open space with vegetation where heath was an important component. In addition to heath grasses also formed part of the vegetation in the open places. Arboreal pollen percentages fluctuate between barrows from around 45 to around 75 %, indicating varying sizes of the open spaces the barrows were situated in. Based on research that was performed in recent heath areas with varying distances to the forest, such arboreal pollen percentages indicate that the open spaces had an average distance to the forest from 30 until 250 m (Doorenbosch, 2013, chapter 7). The forest consisted mainly of oak (*Quercus*) and hazel (*Corylus*), while alder carr (*Alnus*) was present in the wetter parts of the landscape. The barrows that date to the late Neolithic B period show a similar vegetation composition. Apparently these barrows were also built in open spaces, where heath and grasses were the main components. The size of the open spaces seems to be smaller than during the Neolithic A period. Arboreal percentages are lower, indicating an ADF (average distance to the forest) of approximately 50 m (Doorenbosch 2013, chapter 7).

Besides the barrows that have been palynologically investigated in this area, palynological data from many other barrows on the Pleistocene cover sand areas in the Netherlands are known (see Sects. 3.2, 3.3; see also Doorenbosch, 2013). These data show that these barrows were also built in heaths. Only a fraction of the barrows has nowadays been preserved (Bourgeois, 2013, p. 40), originally the number of barrows was much higher in the Netherlands, and only a part of these preserved barrows have been palynologically analysed. Considering that all investigated barrows were built in heath vegetation it is probable that the non-investigated barrows in these areas were also built in open spaces where heath vegetation was dominant. The barrows of Renkum were built in an alignment and the distance between the barrows is mostly less than a few hundred meters. Since the *average* distance to the forest of the open spaces varies from 50 to 300 m, it is likely that the open spaces were connected to each other, forming relatively small but long-stretched heathland areas with a length that could add up to several kilometres (Fig. 5).

**3.2 Vaassen–Niersen**

A second example is given in Fig. 6, showing the results for several barrows that are situated in the northeastern part of the Veluwe (for the location see Fig. 2). In this area several barrow alignments and solitary

barrows are present. Palynological data from the old surfaces underneath barrows, sods from several periods in which the barrow was constructed (in several cases barrows were constructed in multiple phases). The barrow was reused for secondary burials and new sods were added to the original barrow to cover these burials.as well as from ditches associated with the barrows are available for five barrows in this area. These barrows were dated from the late Neolithic A to the Middle Bronze Age period (≈1800–1100 BC) (Casparie and Groenman-van Waateringe, 1980; Doorenbosch, 2013). Two of these barrows form part of a barrow alignment. The pollen spectra show that the barrows in this area, like the barrows in the Renkum stream valley, were built in open places with heath vegetation, surrounded by oak forest and alder carr in the lower parts of the area. The open spaces were probably larger than in the Renkum stream valley, with an ADF of around 100 m for the barrows of Vaassen and an ADF of 100–200 m for the barrows of Niersen. The vegetation of the open space seems stable, since the barrow spectra from all represented periods show similar vegetation patterns: an open place with species-poor grassy heathland surrounded by oak forest with an alder carr nearby. Fig. 7 shows the visual impact on the landscape in the area of Vaassen–Niersen, assuming all barrows were built in heath.

**3.3 Echoput**
A third example concerns the twin barrows of the Echoput (site indicated in Fig. 2), which date to the Middle or earlier Late Iron Age (Bourgeois and Fontijn, 2011, p. 87; Doorenbosch, 2013; van der Linde and Fontijn, 2011, p. 62). These barrows were excavated and sampled for pollen analyses by the Archaeology department of the University of Leiden (Doorenbosch, 2011). The pollen analyses were performed by the first author of this article as part of her PhD research (Doorenbosch, 2013). From both burial mounds samples were taken from the old surface and several sods. In addition, the soil profiles underneath both barrows were sampled. Results are shown in Fig. 8, 9, and 10.
The pollen spectra from the old surfaces and sods consist mainly of herbaceous pollen, dominated by heather (*Calluna vulgaris*) and less, but still in considerable amounts by grasses. This indicates that the Echoput barrows were both built in an open space dominated by heather. The open spaces were surrounded by forest vegetation, namely oak, hazel and alder. The open heath areas were probably much larger than at the older barrows described in the first two examples. At the Echoput the ADF was around 200–300 m. The open spaces were not recently created before the barrows were built. The heath vegetation had already had time to establish and to develop and the open place must have existed at least some years before the barrows were built. This is confirmed by the pollen diagrams shown in Fig. 9 and 10. These diagrams show the vegetation development from a certain period prior to the barrow building. Although the soil profiles have not been dated, it is clear that heath was already present some time before the barrows were constructed. The diagrams show a decrease of the surrounding forest and an increase of the heath vegetation and at the time the burial mounds were built vegetation was dominated by heather.

**3.4 Bedafse Bergen**
The "Bedafse Bergen" is a biogenic land dune ridge western of Rakt, a historical complex of a hamlet and arable fields, surrounded by coppice. West of the hamlet, cattle heathland was present. A plaggic Anthrosol developed on the arable fields (Fig. 11, phase A). After 1000 AD the heaths in this region were threatened by sand drifting due to severe forest degradation. The degradation was caused by complete deforestation during the 11th until 13th century (Vera, 2011; van Mourik et al., 2012a,b). This deforestation triggered the first regional extension of (older) sand drifting; aeolian eroded sand was transported by the southwest winds from the heaths to Rakt. The coppice hedge around the hamlet served as protecting screen and initiated the building of a ridge of inland dunes, the Bedafse Bergen (Fig. 11, phase B). The introduction of the deep stable economy in the 18th century (Vera, 2011) initiated the second extension of (younger) sand drifting (Fig. 11, phase C). At the east side of the ridge, the western edge of the plaggic Anthrosol was buried by (younger) drift sand, at the west side of the ridge we could excavate the Podzol in cover sand, buried by older drift sand. Both profiles were sampled for pollen analysis and optically stimulated Luminescence (OSL) dating (table 1 and 2). The Ah of the buried Podzol in profile Bedafse Bergen was also sampled for radiocarbon ($^{14}$C) dating (table 2). The OSL datings have been performed in the NCL (Wageningen University), the radiocarbon dating (accelerator-mass-spectrometer) in the CIO (Groningen University); the methodology of the dating techniques has been described in van Mourik, 2012a). The development of the landscape around the Bedafse Bergen is presented in Fig. 11, the sampled profiles in Fig. 12. The pollen diagram Rakt is shown in Fig. 13, pollen diagram Bedafse Bergen in Fig. 14. During the transformation of heaths into new arable land and forests in the 20th century, the majority of the soil archives has been destroyed. For this reason, the soil archives of the Bedafse Bergen have a high scientific value. The position of the sampled profile is indicated in Fig. 11c. The oldest formation is cover sand. The post-sedimentary infiltrated pollen spectra of the 3Ap in cover sand demonstrate that arable land was

created on *Calluna* heath. Conventional radiocarbon dates of the humin and humic acid fractions of similar profile on the nearby Maashorst indicate a start of sedentary agriculture around 1000 BC (van Mourik, 2012a). The OSL dating (L1) of the 3Ap reflect ploughing of the agricultural soil until around AD 100 (ploughing resulted in bleaching of quartz grains, originally part of the cover sand deposit). Until ≈AD 1600 the plaggic deposition rate was relatively low (2Aan2). After AD 1600 the OSL datings point to an acceleration of the plaggic deposition (2Aan1), related to an increasing content of mineral grains of the plaggic manure. It is known that in the course of the 18th century farmers used, in addition to straw, *Calluna* heath sods as stable filling (van Mourik et al., 2016). The difference in acceleration rate is also reflected by the pollen density curve. The plaggic Anthrosol was buried by (younger) drift sand around AD 1800.

The buried Podzols of the profiles Bedafse Bergen and Rakt developed originally at the same stratigraphic level in cover sand. The position of the sampled profiles is indicated in Fig. 11c. The Podzol Bedafse Bergen is buried below 12 m high dune sand deposits. Only the basic layers of the drift sands and the top horizons of the Podzol have been sampled for pollen analysis and dating of the basis of the older drift sand deposits. The pollen spectra of the horizons of the buried Podzol contain high percentages of Ericaceae, pointing to the presence of heath. The relatively high percentages of *Quercus* and *Corylus* refer to deciduous forest in the surroundings. The pollen spectra of the drift sand layers show lower percentages of Ericaceae and higher percentages of Poaceae, indicating heath degradation. The decrease of *Quercus* refers to deforestation. The oldest drift sand layers have been deposited between ≈AD 1300 and ≈AD 1500. This correlates to a period of deforestation, as has been documented in historical documents (Vera, 2011). The radiocarbon age of the humic acids, extracted from the 3Ah is ≈AD 725. It is known that radiocarbon ages of humic horizons of palaeosols overestimate the real ages (van Mourik et al., 2010).

**4 Discussion**
**4.1 The barrow heath landscape and heath management**

[revised manuscript text omitted]

**4.2 Heath in the Roman Period**

Unfortunately, not much is known about heath and the maintenance of heath during the Roman Period. The practice of barrow building was no longer continued and no soil records have been investigated that could give information on the use of heathland in the period thereafter. Some studies have been conducted, however, suggesting that heathland areas continuously existed throughout the Roman Period, indicating that some form of heath management must have taken place during that time, for example at the Echoput (see Sect. 3.3). At the Echoput heath was present at and prior to the period the barrows were built (the late Iron Age). This heath area must have been managed for some time to maintain. At the same site several post holes have been discovered close to the barrows. The infill of these post holes, which probably date to the Late Medieval Period, have been analysed for pollen as well (Doorenbosch 2013, chapter 8.1). It was shown that at the time the posts were placed the heath had expanded compared to when the barrows were built. It cannot be said with certainty that the heath was present and maintained in the period in between the barrow building and the post placing, but it is likely that this is the case. The presence of heath during the Roman Period was also shown in pollen diagrams from several palaeosols, for example Venloop (Maashorst, North-Brabant, van Mourik et al., 2012a) and Defensiedijk (Weert, Middle Limburg, van Mourik et al., 2010). In addition, the presence of heath during the Roman period is mentioned in several other studies (Bakels, 2014; Kooijstra, 2008; Kooijstra and Groot, 2015). With the continuous presence of heath vegetation it is most probable that the local population continued with extensive grazing management on the heaths until the early Middle Ages. In Twente and Drenthe (Eastern Netherlands) the sustainable management of the common heaths was controlled by the so called marke administrative system (Spek, 2004), in the Campina (Southern Netherlands) the stability of the common heaths was the common denominator of sustainable agricultural production of the smallholders and the elites (de Keyzer, 2013). Small scale events such as storms and fires could destabilize the ecosystems for a relatively short time, but commercial interventions caused more permanent destabilization and sand drifting: the iron industry from the Iron Age till the 7[th] century (Beukenkamp & Sevink, 2005), the deforestations from the 11[th]-13[th] century and the introduction of the deep stable economy in the course of the 18[th] century (Vera, 2011).

**4.3 Heath management since AD 500: plaggic agriculture, sand drifting, reclamation and restoration**

In the Early Middle Ages people learned to produce manure by using organic materials such as the ectorganic horizon of deciduous forests and grass sods from the brook valleys as stable fillings (van Mourik and Jansen, 2016). Serious degradation of heaths during this time is not recorded in the soil archives. Heath management was most probably restricted to burning and mowing of older *Calluna* shrubs (Burny, 1999).
During the 11[th] until 13[th] century landowners cut the last forests, because the prices of wood were going up (Vera, 2011). These deforestations resulted in regional extension of sand drifting (van Mourik et al., 2012a,b). To acquire fuel, farmers dug sods of the ectorganic horizon of the moist heaths (Burny, 1999), but after the removal of the ectorganic sods, the moist heaths will recover in two until four years and sand drifting was not

an issue. From archived documents it is known that the farmers protected the dry heaths carefully against sand drifting (Vera, 2011). But in the course of the 18th century the combination of population growth and increase of food demand resulted in the extension of arable fields and the increase of the production of stable manure. Lack of stable fillings was compensated by the use of humic sods from the dry heaths (Vera, 2011; Burney, 1999). Mowing and burning were sustainable management rules, but sod digging caused degradation and initiated the next wave of sand drifting (van Mourik et al., 2012a,b).

At the end of the 19th century, the plaggic agriculture came to an end. Due to the combination of the fall of prices of wool and the introduction of chemical fertilizers and urban compost the heaths lost their economic value and the government started with the reclamation of the heaths into new arable fields. After the introduction of hybrid maize in 1950 AD, these fields became the base of the bio industry (chicken, beef and pork). The heathlands that were damaged by sand drifting were turned into forests, mainly Scotch pine plantations. Figure 2 shows how the heath surface diminished from ≈600 kha in 1850 to ≈30 Kha in 1990. Since 1980 the government started the development of the national ecological master structure to improve the biodiversity on national and regional scale. The program Natura 2000 included the preservation of the remaining heaths and restoration of lost heaths. As a result, the heath area had increased from 30 to 35 kha in 2008. Most of these heaths are now parts of protected nature reserves, in which also the (last) valuable soil archives such as barrows, buried Podzols and plaggic horizons are protected as elements of the geological and cultural heritage.

Due to the presently increased nitrogen concentrations in rain and ground water, heaths cannot survive without management measures to prevent an accelerated succession to brushwood and forest. Applied measures are, like in the 19th century, intensive grazing, mowing, burning and sod digging (Smits and Noordijk, 2013).

**5 Conclusions**

– The invasion of heaths into the cover sand landscape of the Netherlands is associated with forest degradation; first at small scale in open places in the original forest, which existed due to natural causes, followed by larger scale anthropogenic deforestations.

– People created and maintained the barrow landscape on ancestral heaths from the Late Neolithic until the Late Iron Age.

– For these people, heath was not only economically valuable, but also culturally.

– During and after the Roman Time, people continued with heath management, mainly by cattle grazing; the heaths maintained their economic, but lost their cultural value.

– Introduction of the plaggic agriculture system around AD 500 resulted in further soil acidification of the heaths and the development of plaggic horizons on arable fields.

[revised manuscript text omitted]